# Ground-Based Experiment for Electric Propulsion Thruster Plume—Magnetic Field Interaction

Andreas Neumann [1],[*] and Nina Sarah Mühlich [2]

1    German Aerospace Center, DLR, 37073 Göttingen, Germany
2    ArianeGroup, Orbital Propulsion, 74239 Lampoldshausen, Germany
*    Correspondence: a.neumann@dlr.de

**Abstract:** Electric space propulsion is a technology which is employed on a continuously increasing number of spacecrafts. While the current focus of their application area is on telecommunication satellites and on space exploration missions, several new ideas are now discussed that go even further and apply the thruster plume particle flow for transferring momentum to targets such as space debris objects or even asteroids. In these potential scenarios, the thruster beam impacts on distant objects and subsequently generates changes in their flight path. One aspect which so far has not been systematically investigated is the interaction of the charged particles in the propulsion beam with magnetic fields which are present in space. This interaction may result in a deflection of the particle flow and consequently affect the aiming strategy. In the present article, basic considerations related to the interaction between electric propulsion thruster plumes and magnetic fields are presented. Experiments with respect to these questions were conducted in the high-vacuum plume test facility for electric thrusters (STG-ET) of the German Aerospace Center in Göttingen utilizing a gridded ion thruster, an RIT10/37, and a Helmholtz coil to generate magnetic fields of varying field strength. It was possible to detect a beam deflection on the RIT ion beam caused by a magnetic field with an Earth-like magnetic field strength.

**Keywords:** electric propulsion; ion beam; magnetic field; vacuum test facility; space debris

## 1. Introduction

The interest of commercial industries in electric space propulsion did a leap forward triggered by Boeing's all-electric initiative based on its new 702SP platform [1]. Electric propulsion (EP) is nowadays used routinely for satellite station-keeping, and recent developments are using EP for a complete orbit transfer. Electric propulsion is also gaining more interest in the sector of future science missions requesting very low thrust in conjunction with low thrust noise and accurate thrust level control [2].

EP systems generate low absolute thrust compared to thruster mass, but the propellant efficiency is significantly higher compared to conventional chemical thrusters. The high propellant efficiency opens the door to new applications which would not be possible with conventional systems. One application is non-contacting space debris removal using ion beams as it was investigated in the project LEOSWEEP. This is a project funded by the European Union within its Framework Programme 7 involving 11 partners [3]. Its goal was to investigate the possibility of non-contacting space debris removal using ion beams. A tractor satellite approaches the debris object up to several tens of meters and fires its ion engine onto the object. The beam impact causes a force that changes the momentum of the object, which will change its orbit. The concept was also presented in 2011 in [4]. A similar idea for repositioning space objects is discussed in [5].

Such a concept might also work on larger objects such as asteroids, an approach that is outlined in [6]. Figure 1 shows both these concepts. In (a), a shepherd satellite equipped with two opposite-acting thrusters changes the momentum of a space debris object. Here,

an obsolete rocket upper stage is considered. In (b), a shepherd satellite is utilized for momentum transfer onto an asteroid. The resulting change of its path of motion may prevent an impact of the asteroid on Earth.

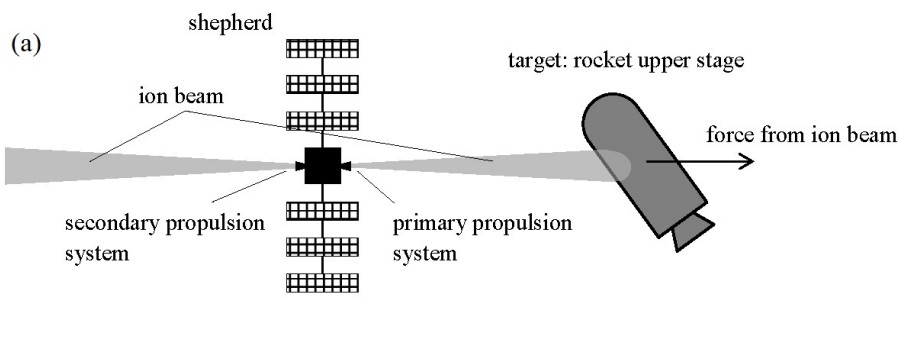

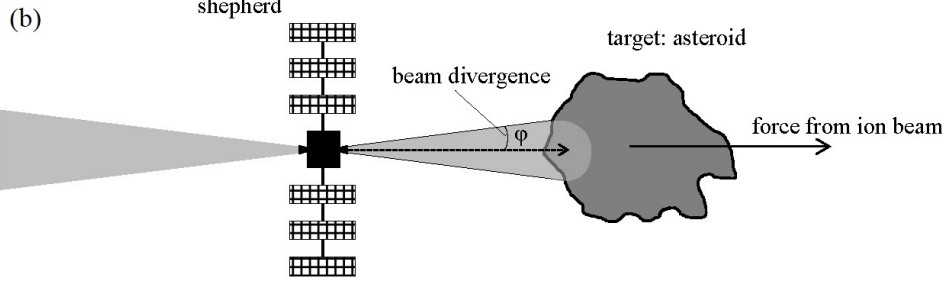

**Figure 1.** Principle potential momentum transfer scenarios on space objects. (**a**) Shepherd satellite changes the momentum of a space debris object, i.e., an obsolete rocket upper stage. (**b**) Shepherd satellite induces momentum transfer on an asteroid (picture: DLR).

In both these concepts, the ion or plasma beam has to be directed onto objects at distances of several tens of meters or even more. The aim is to move the object, giving it a well-defined moment. If the beam push is not directed through the center of mass, an additional angular momentum will be given. Such an effect is unwanted, and this constellation poses high demands on beam collimation and on beam pointing accuracy. In order to give a rough number, one should be able to hit the object with an accuracy of 1% of its size, e.g., 10 mm for a 1 m object. This implies that all possible sources of ion beam deflection and potential errors must be characterized and quantified prior to the mission definition.

Furthermore, the impact of an ion beam on a target must be known, e.g., sputtering effects, which can also cause an interaction with the shepherd satellite itself.

In the present work, the emphasis is put on the interaction of the thruster beam with magnetic fields which are present in the space environment.

The difference to a classical well-defined narrow and unidirectional ion beam (such as in ion optics or particle accelerators) is that we have mixed ion velocities, a not negligible divergence angle, and even some electrons, e.g., from secondary processes. One trigger for this paper was the disagreement in the EP community about the effects of a magnetic field on a typical electric thruster beam.

A few studies have been performed on this subject. In [7], Cichocki et al. compared a particle in cell formalism with a fluid formalism for investigating a thruster's space plume under different conditions.

Korsun et al. described the expansion and propagation of plasma clouds in [8]. They called such a plasma "plasma artificial formation" or PAF. In [9], a more detailed analysis of a beam in magnetic fields is described, and an anisotropic beam shape is mentioned. Korsun et al. carried out more work on modeling the plume behavior in [10]. They used a 1 kW Hall-effect thruster and a longitudinal magnetic field of 0.2 Gauss (20μT), which they attribute to be present in 1000 km altitude. This magnetic field, if perpendicular to the thruster flow axis, is producing a compression in the plane perpendicular to the magnetic

field and hence confirmed the anisotropy stated in the past. On the other hand, such effects are not always accepted in the space propulsion community. Therefore, we decided to conduct first experiments concerning the impact of magnetic fields on EP space thrusters and to show what is feasible in ground testing with respect to such a subject.

At the electric propulsion test facility of DLR, experiments with beam shape analysis and beam propagation at different distances have been performed. Some of these experiments were carried out within the German project RITSAT and within the above-mentioned European Commission co-funded project LEOSWEEP.

The work within the project RITSAT had the goal of bringing together engineers, materials scientists, chemists, and physicists for working together on aspects of EP technology and in particular on developing the radiofrequency thruster (RIT) technology, ranging from μN-RIT to large RIT-35 thrusters. The project included new developments of plasma and plume diagnostics and their usage in the test facilities JUMBO in Giessen and STG-ET Göttingen [11]. Another aspect was the investigation of thruster/spacecraft interaction in test chambers and the complementary electronics and plasma modeling [11].

## 2. Thruster

One of the thrusters used by DLR in the above-mentioned projects is the radiofrequency ion thruster RIT10/37. The working principle is shown in Figure 2. The number "10" stands for the grid diameter in cm, and "37" stands for the number of holes in the grids. For this type of thruster, the ionization is realized via a high-frequency field emitted by an antenna coil outside of the discharge chamber. Electric fields between the grids accelerate the ions up to an exhaust velocity of several tens of km/s [12].

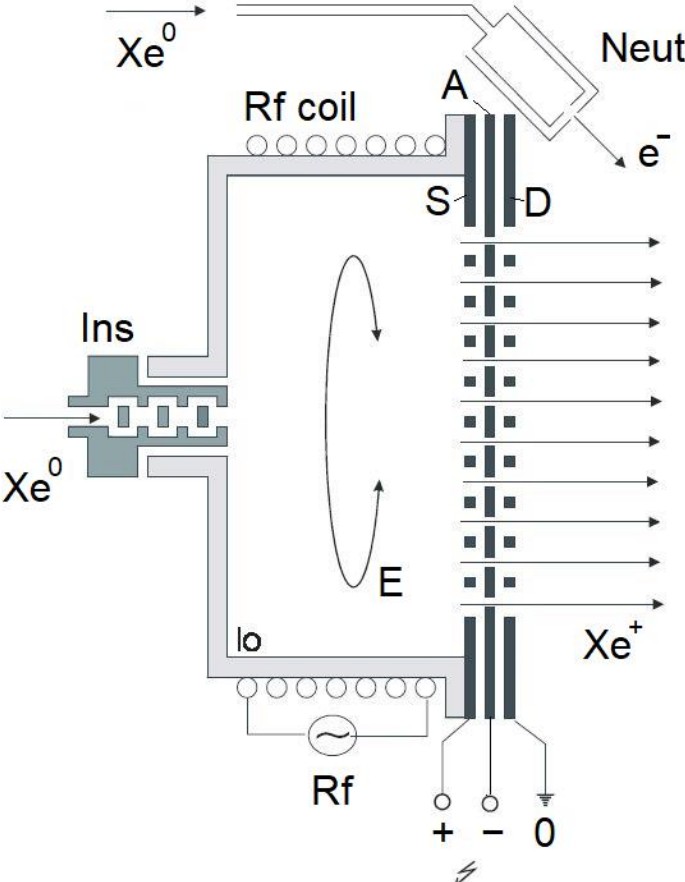

**Figure 2.** Principle of a radio frequency ion thruster, RIT. Io: ionization chamber; Rf: radiofrequency generator; Neut: neutralizer; S: screen grid; A: accelerator grid; D: decelerator grid; Ins: gas inlet [12].



This thruster was part of a couple of EP investigations, especially plasma beam characterization. The thruster can run on several noble gases, but for the experiments performed in this study, xenon was used as propellant.

The thruster is well suited for applications that need a narrow beam because its beam divergence is about 8° degrees, which is small compared to other EP technologies. Another advantage of this thruster is its very stable operation over many hours or even days.

The challenge in the present study was to characterize the deflection of ions generated by an electric propulsion thruster operating at a low beam power of about 50 W and interacting with magnetic fields in the µT range. This requires the measurement of deflections in the range of several mm using a beam scanning device at several meters distance.

The goal of the present study was to conduct a first investigation of the influence of a magnetic field on electrostatic thruster plume deflection by generating magnetic fields with different strengths ranging from the expected field strength in a mission-like space environment to higher field strengths. Based on this approach, the sensitivity of the beam deflection as a function of the applied magnetic field and the amount of the deflection were determined by conduction experiments with and without a magnetic field.

### 3. DLR High-Vacuum Plume Test Facility Göttingen—Electric Thruster, STG-ET

In STG-ET, distances between thruster and target of up to 7 m can be realized. Therefore, this large-scale ground-based vacuum test facility in combination with the available ion thruster and plume diagnostics is well suited for the envisaged fundamental studies of EP ion beam–magnetic field interactions within the range of the above-given boundary data.

The central element of DLR's electric propulsion test facility is a 12 m long and 5 m diameter vacuum chamber. The chamber is mounted on sliding bearings for reduction of mechanical stresses in case of pump-down and temperature changes. The test object and main diagnostics equipment are positioned on a stand which is decoupled from the metal chamber wall. This decoupling ensures less vibrations and a stable and well-defined space coordinate system. Pumping is performed via several turbo and cryopumps. This arrangement is able to reach an operational pressure of around $1$–$5 \cdot 10^{-5}$ mbar, which is a typical requirement for EP thruster operation. For instrumentation and pumping, 169 feedthrough ports are available [13].

Ion thrusters generate beams of fast ions that may interact with the facility walls and equipment. These high-velocity ions cause sputtering when hitting the walls of the chamber or all other components located inside the chamber. In ion propulsion test chambers, dedicated beam dump targets are used for reduction of sputtering effects. The STG-ET beam dump consists of water-cooled stainless-steel plates coated with graphite foils which have a low sputtering coefficient. In addition to the minimization of sputtering effects, the beam dump must be able to take away the heat load generated by EP thrusters, including more powerful ones. In STG-ET thrusters, generating up to 25–50 kW can be operated.

### 4. Beam Scanning Device LDBS

For testing electric propulsion systems, several diagnostic systems are available. Besides thrust measurements monitoring of the ion beam distribution in space is an important task, several systems are available in STG-ET, such as scanning arms equipped with, e.g., Faraday cups or retarding potential energy analyzers (RPA). The diagnostic system used for this experiment is the long-distance beam scanner (LDBS), which is able to scan a flat area of 3 m × 1 m at a distance of about 2–8 m from thruster exit [14]. Figure 3 displays the schematic of the LDBS system. The sensors are two Faraday cups installed for redundancy.

The LDBS absolute accuracy in space coordinates is estimated to be ±3 mm, and the reproducibility is <±1 mm, based on our own measurements, as this system is fully custom made. The sensor heads are Faraday cup detectors. One comes from Kimball Physics, FC70, and has a 1.6 mm aperture diameter. The other was manufactured by the

University of Giessen, Germany, and has a 5 mm aperture diameter. For none of them, absolute measurement errors are given. The current signal was recorded by a dual channel Keithley Picoamperemeter with an accuracy of 0.03% + 5 nA in the current range. As for the beam movement, relative measurements are most relevant, and the overall accuracy in intensity is estimated to be <1%.

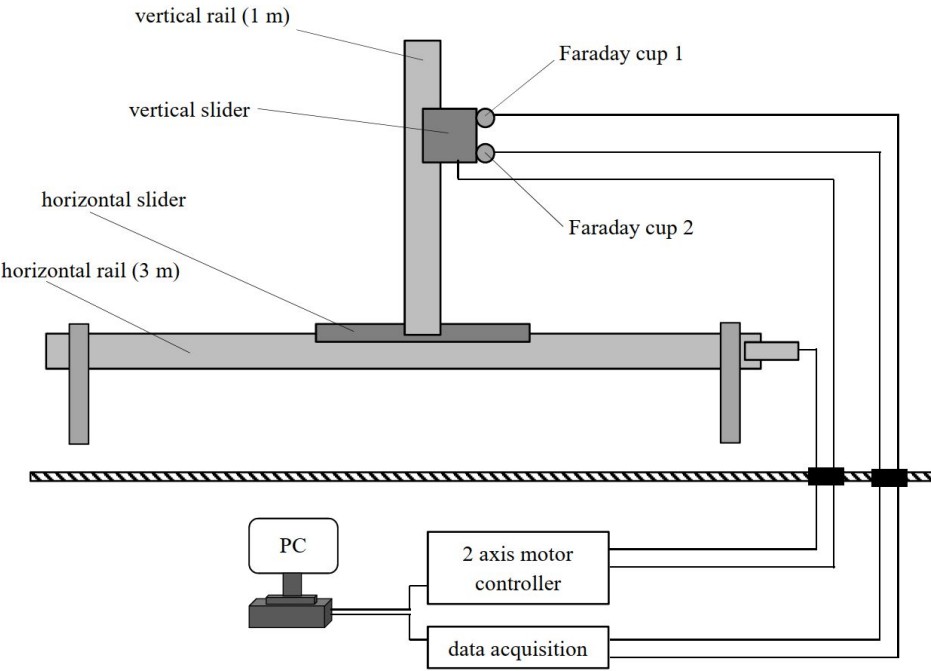

**Figure 3.** LDBS measurement system: The flat field scanner is able to map an area of 3 m × 1 m. Two Faraday cup sensors are used for redundancy (see text).

A typical 2D scan is recorded in about 15–45 min, depending on frame and step size settings. Since the utilized thruster generates a stable beam, no influence of the scan recording times on the measured beam deflection is expected.

For this study, the LDBS was positioned at a distance of 3.85 m from the thruster exit. This distance could have been made longer, but then the beam would have been larger and the density of ions lower, which results in longer scan times.

## 5. Magnetic Field Environment

In case of space debris removal missions operating in altitudes typical for low Earth orbits (LEO), the main magnetic field source is the Earth's magnetic field. Its field strength is variable in space and time, and only a few representative values are given here. Computational models for the magnetic field exist, e.g., the NOAA tool [15]. Furthermore, in situ measurements are available, e.g., [16].

In a first step, the local magnetic field is calculated using the NOAA tool. Table 1 lists the Earth's magnetic field at the Göttingen DLR site as calculated herewith. The total field intensity is about 49 μT, and the vector inclination is 67 degrees. An interesting question is what happens to the local field in the inside of the vacuum chamber STG-ET. For this purpose, measurements were taken at the DLR site inside and outside of the vacuum chamber in the timeframe October–November 2012. The measurements had to be taken manually (ladder, scaffolding), which took some time. Table 2 shows the recording, and the values are slightly lower than the model data but do agree within less than 20%. The real absolute value is not of importance for the investigation within the present study. However, it is important to note that the field vector still has the same direction inside the metal chamber but has a slightly lower absolute value.

**Table 1.** Earth's magnetic field from NOAA model [14].

| Model Input/Output | |
|---|---|
| Latitude | 51.53 deg N |
| Longitude | 9.93 deg E |
| Elevation AMSL | 150 m |
| Model Date | 21 September 2015 |
| Magnetic Field | |
| Horizontal Component | 19.2 μT |
| Vertical Component | 45.2 μT |
| Inclination | 66.9 deg |
| Total B | 49.1 μT |

**Table 2.** Earth's magnetic field measured at DLR Göttingen; field strength outside and inside of STG-ET vacuum chamber.

| | Horizontal Component | Vertical Component |
|---|---|---|
| Outside vacuum chamber | 16.0 μT | 42.6 μT |
| Inside (open) vacuum chamber | 16.4 μT | 36.1 μT |

The above field intensities are for a location on the ground and not in a space orbit, but the decrease as a function of altitude above ground up to a LEO orbit is not very pronounced. According to the NOAA model, the total value at 300 km is 43 μT and decreases to 33 μT in 900 km (all data taken above the coordinates of Göttingen). Therefore, the simplification for this investigation is to use 40 μT as a representative order of magnitude for the Earth's magnetic field in LEO.

## 6. Experimental Magnetic Field Generation in STG-ET

As stated, the typical Earth magnetic field in the STG-ET is about 40–50 μT. This is the background field permanently present, and this study needs an additional field that adds up to a larger field with a measurable impact on the ion beam.

The additional magnetic field was generated utilizing coils in Helmholtz arrangement. The basic configuration is shown in Figure 4, and the principle can be found in [17]. The sketch shows how the fields of the individual coils add up to generate a homogeneous field inside the space confined by the coils. In the present setup, a modification of the basic Helmholtz arrangement has been chosen, the coils being square and not circular. This approach proved to be easier to manufacture. Other constraints were the size of the vacuum chamber, the distances between the thruster and the diagnostics, and the handling of large components inside the vacuum chamber. This led to a side length of the coils of 1640 mm and a spacing of 814 mm.

As the coils have to operate in a vacuum, the heat load generated by the coil current had to be carefully assessed, and a maximum current of 5 Amperes was chosen. Combined with the other coil geometry data, this leads to a maximum field strength of about 250 μT in the ideal configuration, which is a factor of five higher than the Earth's magnetic field (see Section 5).

After assembly of the coils, the magnetic field was measured at different points inside the coil volume in order to confirm the specifications. Figure 5 shows the result of nine different measurement points inside the coil, each of them being an average over three measurements at different heights. The dip from edge to center is about 15 μT, and the average of all measurements leads to a field strength of 235.9 μT. The field should change in a linear manner with the applied current, and this is confirmed and shown in Figure 6, which compares measurements and Helmholtz coil calculations. In Figure 6, only the field center intensity is used. The linear dependence can be obviously seen.

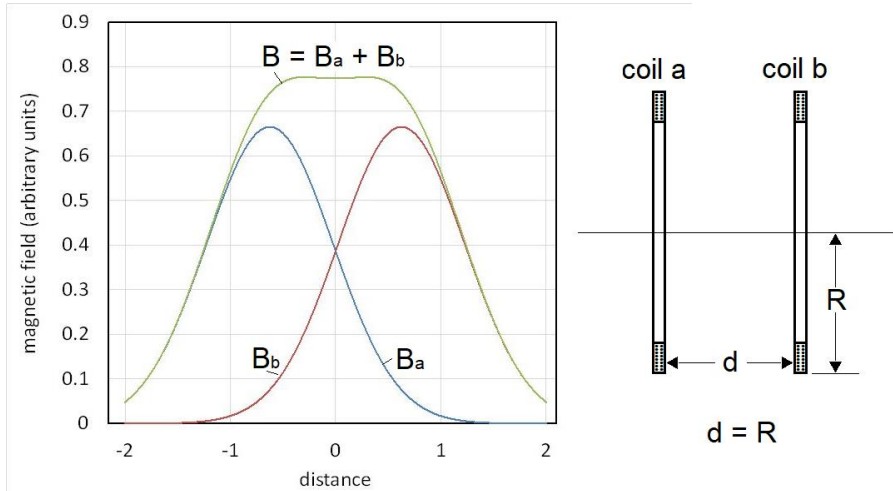

**Figure 4.** Principle of Helmholtz coils scheme used to generate a homogeneous magnetic field. The spacing d of the two coils is equal to the radius R of the coils, and the magnetic field is homogeneous in the space between the coils picture: DLR ).

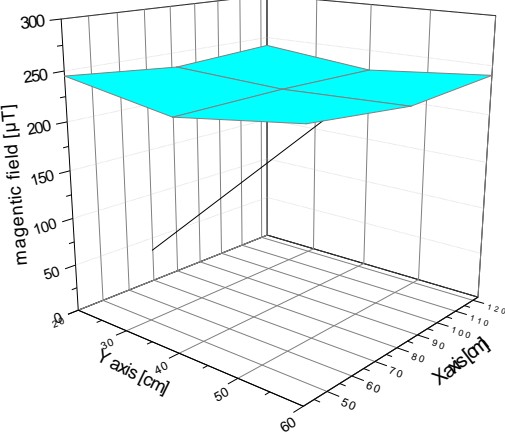

**Figure 5.** Measured magnetic field distribution inside the coils.

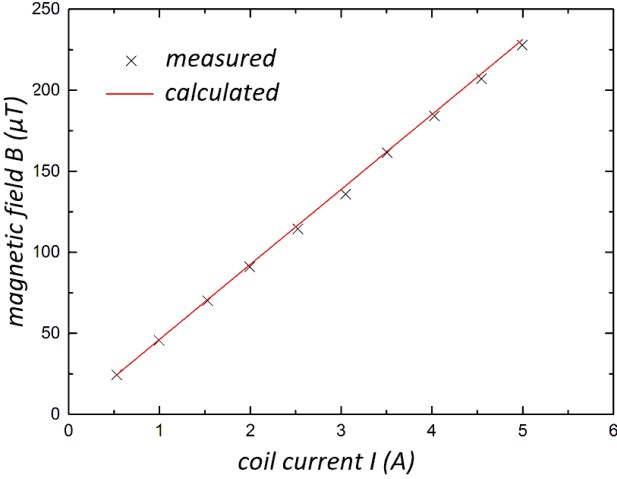

**Figure 6.** Measured magnetic field strength at the center of the Helmholtz coil compared to the calculated field. The maximum deviation is less than 3.6%.

### 7. Calculation of Ion Deflection

With the above coil specifications, it is possible to assess a deflection of ions under various conditions. The principle of the experimental setup is shown in Figure 7. It includes the ion-generating source, a coil located in the flight path, and an observer or detector system.

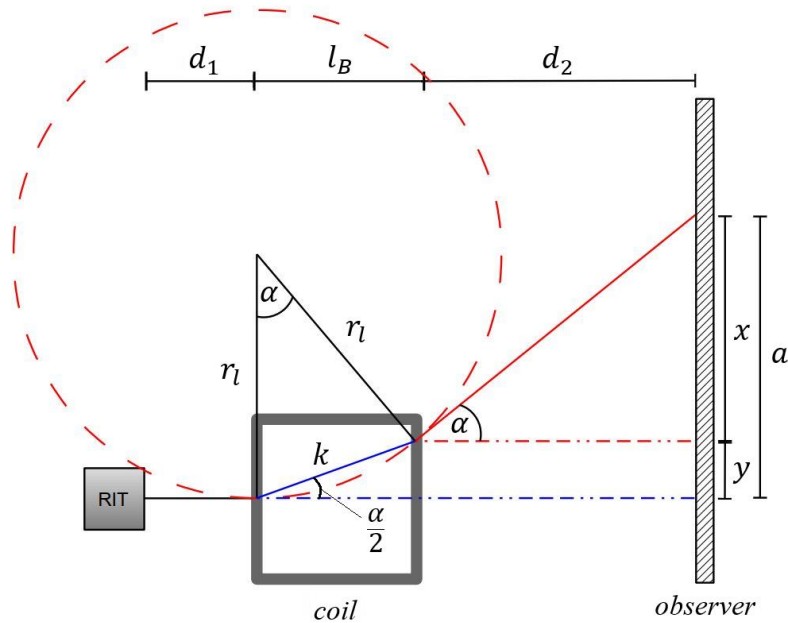

**Figure 7.** Sketch of the geometry used for calculating the deflection of an ion path in a magnetic field; the magnetic field is perpendicular to the image plane and generated by the square coil.

The magnetic field is perpendicular to the image plane and generated by the square coil. The distances between thruster and coil $d_1$, coil and beam analysis system $d_2$, and the square coil size $l_b$ are:

$$d_1 = 960 \ mm$$
$$d_2 = 1250 \ mm$$
$$l_b = 1640 \ mm$$

It is a well-known fact that the movement of a charged particle in a magnetic field is governed by the Lorentz force (e.g., Feynman [18]). Without additional forces, the perpendicular magnetic field forces an ion into a Larmor circle with radius $r_l$ given by

$$r_l(v, m, q, B) = \frac{mv}{qB} \tag{1}$$

Here, the particle has a mass $m$, velocity $v$, charge $q$, and travels in a perpendicular magnetic field $B$. The initial velocity is determined by the ion source, here an EP thruster (see Figure 2). This velocity can be calculated from the thruster acceleration voltage (RIT) by

$$\frac{m \cdot v^2}{2} = q \cdot U \tag{2}$$

or

$$v = \sqrt{\frac{2 \cdot q \cdot U}{m}} \tag{3}$$

For example, using data for xenon ions with a single charge and a typical acceleration voltage of 1800 V, a velocity of about 52 km/s is obtained. Subsequently, the Larmor radius can be derived from Equation (1).

A simple theoretical approach to pre-estimate deflections in the experimental setup and in near-Earth space is presented here. Several simplifying assumptions are made. In

the first place, without a working neutralizer, the beam only contains positively charged xenon ions.

A further simplification is to assume the magnetic field to be constant inside the coil and zero outside, as with ideal Helmholtz coils. The charged particles interact only inside the coil volume with the field and are in free flight upstream and downstream. Inside the coil area, the magnetic field forces the ions into part of a circular path.

Let us first focus on the magnetic field region inside the coil (Figure 7) and deduce the shift caused by the field on a charged particle entering the field from the left and exiting the coil on the right, travelling along a Larmor radius $r_l$. For a circle chord we have:

$$k = 2 \cdot r_l \cdot \sin\left(\frac{\alpha}{2}\right) \tag{4}$$

Using the coil side length $l_B$, we obtain:

$$cos\left(\frac{\alpha}{2}\right) = \frac{l_B}{k} \tag{5}$$

By application of trigonometry calculus, we eliminate the sine/cosine and obtain solutions for $k^2$:

$$k^2_{1/2} = 2 \cdot r_l{}^2 \pm 2 \cdot r_l \cdot \sqrt{r_l{}^2 - l_B{}^2} \tag{6}$$

Looking at the physical geometry, the boundary conditions state that $k$ must approach zero for $l_B$ going to zero. This leads to the solution:

$$k^2 = 2 \cdot r_l{}^2 - 2 \cdot r_l \cdot \sqrt{r_l{}^2 - l_B{}^2} \tag{7}$$

For the displacement $y$ at coil exit, we have:

$$k^2 = y^2 + l_B^2 \tag{8}$$

Combining Equation (7) with Equation (8) leads to the displacement:

$$y = \sqrt{2 \cdot r_l{}^2 - 2 \cdot r_l \cdot \left\{ \sqrt{r_l{}^2 - l_B{}^2} \right\} - l_B{}^2} \tag{9}$$

We can use $y$ in Equation (9) for calculating the deflection in free space with a homogeneous field perpendicular to the ion flight path and extending over the whole trajectory. Looking at the literature discussing ion beam space debris, removal distances of >10 m are assumed. In Ref. [3], a distance of 15 m is used between shepherd and debris object, and in Ref. [4], the distance is 20 m. For the estimates in this study, 20 m is used as a safe distance.

Some results are listed in Table 3, which shows deflections for singly charged ions of xenon and argon, typical EP propellants. The acceleration voltage 1800 V is typical for our RIT, but for a better focused thruster beam 3000 V may be an option [3]. A magnetic field of 40 µT will deflect a xenon ion accelerated with 1800 V by 114 mm after having travelled a distance of 20 m. Going to faster ions (3000 V) will reduce the deflection to 89 mm. Using argon and equal parameters roughly doubles the deflection, due to a smaller atomic mass.

**Table 3.** Ion deflection in space for a perpendicular magnetic field of 40 µT.

| Gas/Ion | Mag. Field Path $l_B$ (m) | Acceleration Voltage (V) | Deflection (m) |
|---|---|---|---|
| Xenon (1+) | 20 | 1800 | 0.114 |
| Xenon (1+) | 20 | 3000 | 0.089 |
| Argon (1+) | 20 | 1800 | 0.207 |
| Argon (1+) | 20 | 3000 | 0.160 |

For a ground-based experiment like we performed in the STG-ET, we have to include distances without additional magnetic field. This is due to the needs for access and mounting of the coil in between the thruster and diagnostic system. In practice, these distances cannot be made zero. As stated above, the distance between thruster and coil is $d_1$ and between coil and beam analysis system is $d_2$.

According to Figure 7, $d_1$ does not induce any offset because of the absence of a magnetic field. On the other hand, the offset will increase, caused by the directional change of the flight vector at the coil exit along $d_2$. After exiting the field volume, the particle will move along a straight path tangential to the Larmor circle and down to the beam diagnostics system. The angle $\alpha$ is reproduced between thruster axis and this path line. Based on this condition, the distance $x$ can be calculated:

$$tan\,\alpha = \frac{x}{d_2} \tag{10}$$

$$x = d_2 \cdot tan\,\alpha \tag{11}$$

The angle $\alpha$ can be calculated from Equation (5):

$$\alpha = 2 \cdot arccos\left(\frac{l_B}{k}\right) \tag{12}$$

According to Figure 7, the total deflection is:

$$a = x + y \tag{13}$$

By combining Equations (9), (11) and (12) into Equation (13), the total deflection can be written as:

$$a = d_2 \cdot tan\left(2 \cdot arccos\left(\frac{l_B}{k}\right)\right) + \sqrt{2 \cdot r_l^2 - 2 \cdot r_l \cdot \left\{\sqrt{r_l^2 - l_B^2}\right\} - l_B^2} \tag{14}$$

This total ion deflection with respect to different magnetic field strengths is shown in Figure 8. Here, the deflection is shown for acceleration voltages ranging from 1000 V up to 3000 V. A voltage of 1800 V and a $B$ field of 250 µT will result in about 19 mm of deflection in the used setup. For a field strength of 40 µT, a deflection of 3 mm can be expected.

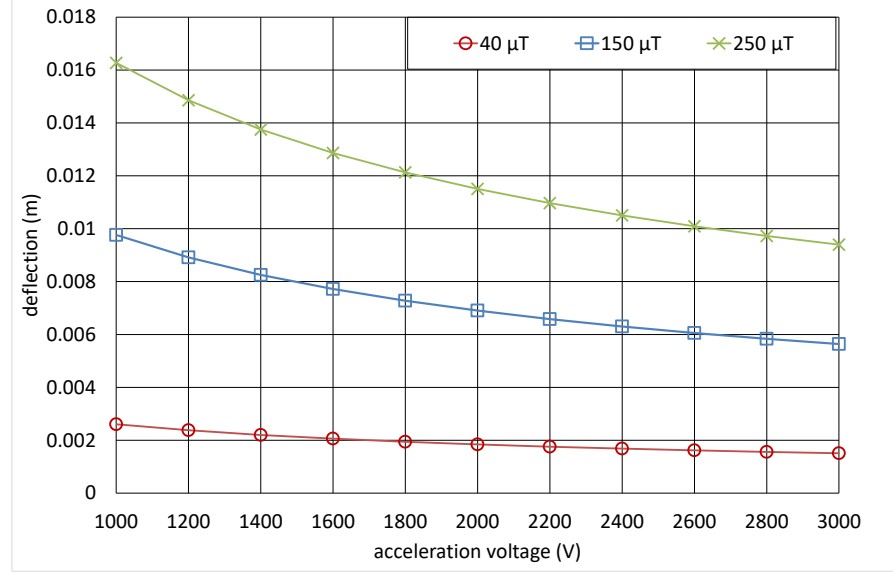

**Figure 8.** Ion deflection for different magnetic field strengths versus ion acceleration voltage. An amount of 1800 V is the voltage used with the RIT thruster for the present study.

As stated above, the thruster ion beam does have a divergence cone of about 8°; therefore, the ions will not all enter the magnetic field under zero angle like displayed in Figure 7. But for the magnetic conditions encountered here, the geometry arguments will be valid, and the divergence should be transmitted to the exit of the beam out of the field.

## 8. Measurement Results and Analysis

The above calculations gave a deflection of about 12 mm for the 250 μT field in the setup of Figure 7. This value is larger than the LDBS resolution and showed that a measurement would give results. On the other hand, a deflection of around 2 mm at 40 μT seemed too small to be measured with the actual setup. Therefore, the approach was to use in the first place the highest possible magnetic field with the Helmholtz coils of 235.9 μT.

A view of the experimental setup inside the vacuum chamber can be seen in Figure 9. The thruster is mounted on the pedestal in the front. The magnetic coil sits between the thruster and the LDBS (structure sitting on the chamber floor behind the ladder).

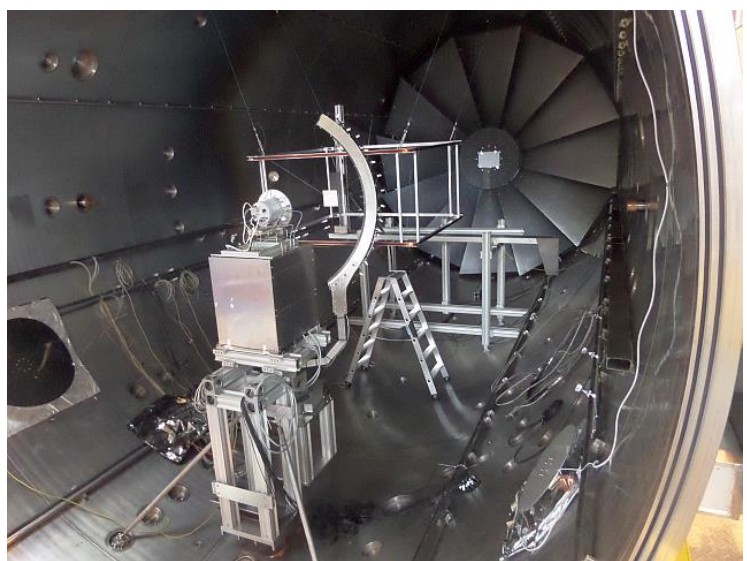

**Figure 9.** View of the experimental setup in the STG-ET chamber: thruster, hanging coils, and LDBS system behind the ladder are visible.

The measurements discussed here were mostly taken with a RIT thruster operation point of 1800 V acceleration voltage and 18 mA beam current, running on xenon gas. An additional beam current of 24 mA was also used in some measurements.

During the measurements, the chamber pressure was in the $10^{-5}$ mbar range. The corresponding mean free path is several meters for xenon, roughly the size of the vacuum chamber.

As stated above, in order to measure the strongest deflection effect, three magnetic coil settings were chosen:

- I = 0 A (no coil current).
- I = 5 A.
- I = −5 A.

Conditions (b) and (c) correspond to 235.9 μT in the up or down field orientation (see Figure 9 for coil position).

A full 2D scan with 18 mA beam current, magnetic field switched off, and taken with the LDBS is depicted in Figure 10. This measurement proves that the beam is well-aligned with respect to the measuring device. The y coordinate (vertical axis) was later only used for checking the thruster alignment and to ensure that the horizontal x profiles are taken across the peak of the beam. The horizontal (x coordinate) beam profile measurements for the different magnetic field conditions are shown in Figure 11.

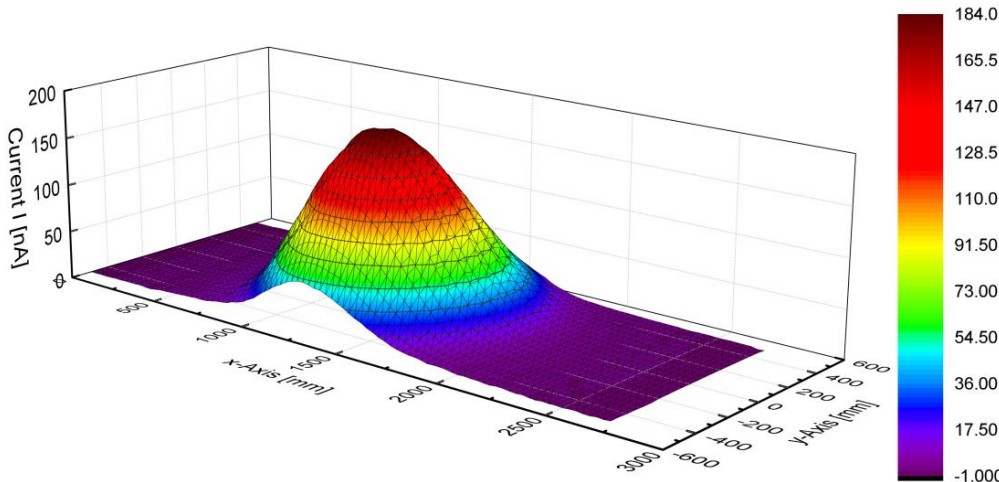

**Figure 10.** Typical beam shape with 18 mA of current. Here, the magnetic field was switched off, but in this 3D representation, the magnetic field impact would not be visible. X is the horizontal coordinate, y is vertical (see Figure 9).

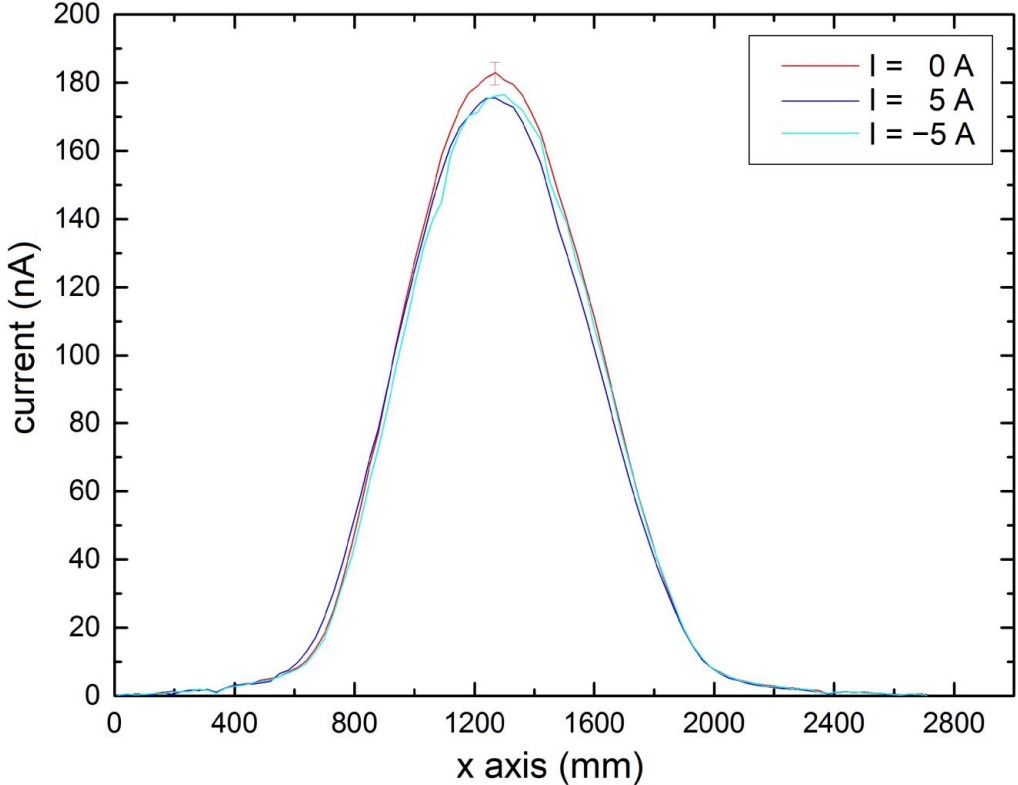

**Figure 11.** Ion current distribution along horizontal x axis at y = 0 for 0 A, −5 A, and 5 A coil current. Beam current is 18 mA (error bar as by LDBS device).

A difference in the curve location between different magnetic field states is visible. In order to determine reliable values for the beam deflection, two independent methods were employed. First, a Gaussian fit procedure was used and second the distribution center of mass method.

A Gaussian function approach is given by:

$$I(x) = I_{offset} + I_0 \cdot e^{-\frac{(x-x_0)^2}{2 \cdot \sigma^2}}$$

(15)

Equation (15) was least-squares fitted to the $x$ profile measurements. $X_0$ is the value that is the fit result, i.e., the center coordinate.

Center of mass (CoM) method is based upon the following formula with a standard deviation (STD):

$$X_s = \frac{\sum(I_i \cdot x_i)}{\sum I_i} \tag{16}$$

$$STD_{X_s} = \sqrt{\frac{1}{(n-1)} \sum_{i=1}^{n} (x_i - X_s)^2} \tag{17}$$

In Figure 12, the Gaussian fit procedure was applied to the raw measured data from Figure 11. Here, the deflection is easier to see. Slight differences in the peak values caused by measurement errors or thruster fluctuations can be eliminated by a normalization. Normalizing the curves to the same peak value and zooming in are shown in Figure 13. The coil powered with +5 A moves the beam to smaller $x$ values ("to the left"), while −5 A deflects the beam to larger $x$ values ("to the right"). The shift direction is in agreement with the basic Lorentz movement assumption.

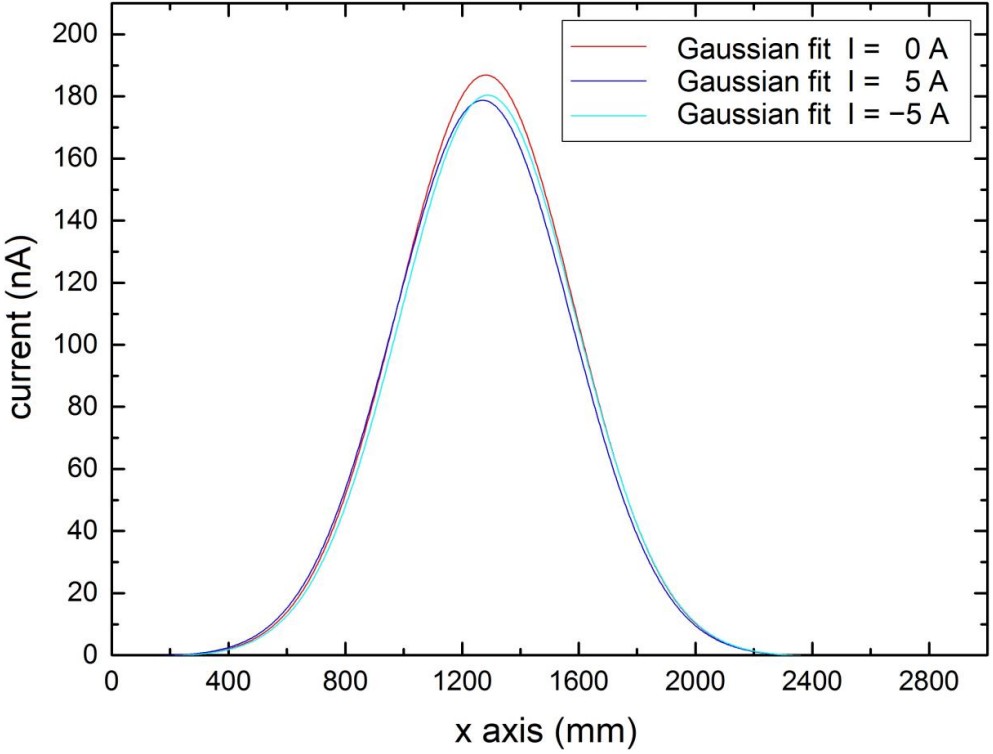

**Figure 12.** Gaussian fit through the data shown in Figure 11.

In order to further confirm the results, an additional test series was performed with a higher beam current of 24 mA. Figure 14 shows a comparison of both the 18 mA and 24 mA measurements using both a Gaussian-fitted beam center and CoM. The beam shape with 24 mA was somewhat different (not shown here), but the beam deflection amplitude was very similar. The error bar sizes mark the standard deviations.

In summary, Table 4 compiles a comparison of calculated deflection versus Gaussian fit center values and CoM results for the thruster operated with 18 mA beam current. The sign (direction) of deflection is correct, and the absolute figures are lower than the calculated values by about 7% for the 5 A current and 52% for the −5 A current (CoM, −6% and −40% for Gaussian fit). The deviation of measurements from calculated values may have different reasons. The magnetic field may not be homogeneous inside the coil volume and may drop at the borders. A smaller deflection would result. On the other hand, the LDBS accuracy in

space coordinates is ±3 mm, but with its better repeatability of 1 mm the impact on the center definition should be minor. It may also be possible that some ion bulk effects cause a modification of deflection compared to basic ion optics.

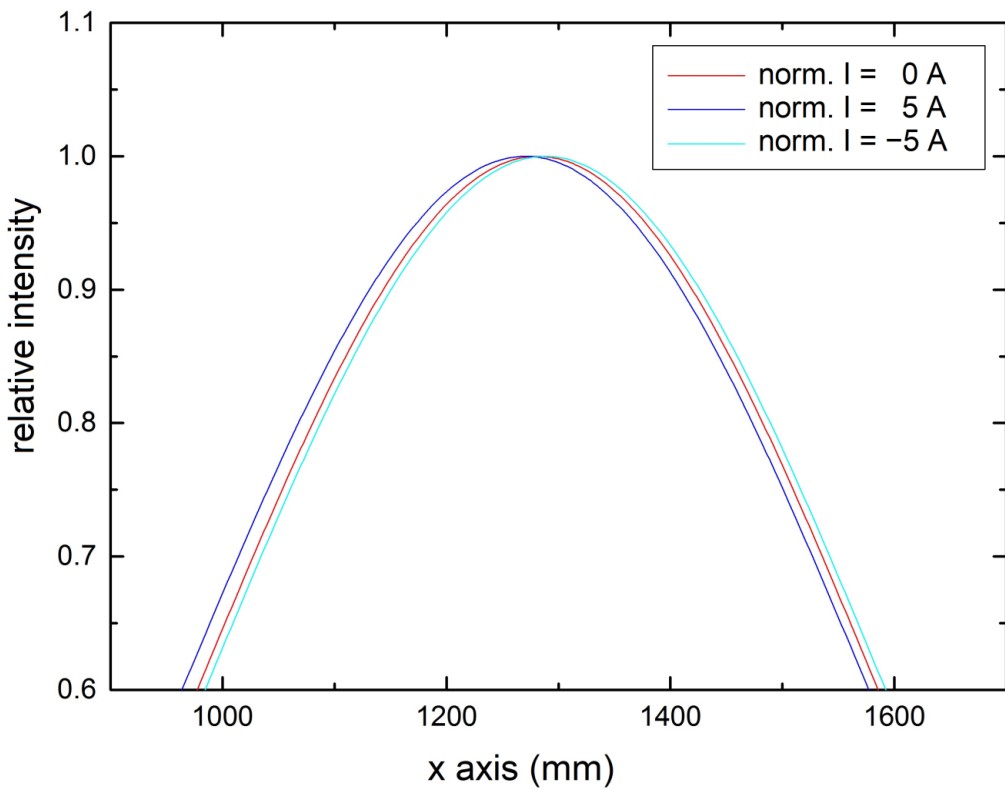

**Figure 13.** Normalized Gaussian fit through the data shown in Figure 11; here, the shift caused by the magnetic field is clearly visible.

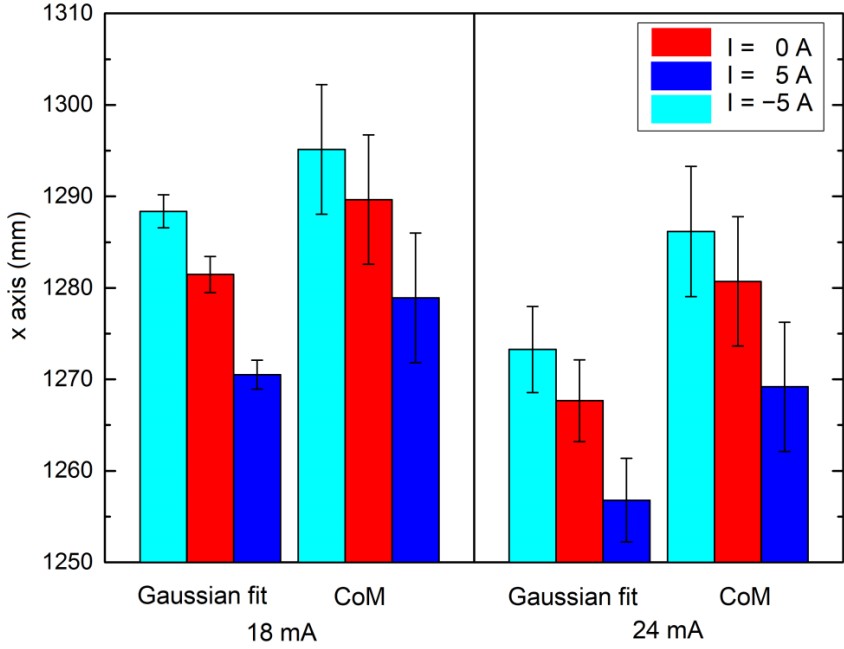

**Figure 14.** Comparison of Gaussian fits and CoM approach for beam movements for 18 mA and 24 mA beam current.

**Table 4.** Comparison of Gaussian fit and CoM for 18 mA beam current.

|  | 5 A $\stackrel{\wedge}{=}$ +235.9 μT | −5 A $\stackrel{\wedge}{=}$ −235.9 μT |
|---|---|---|
| Calculated | −11.45 mm | 11.45 mm |
| $X_S$ from CoM | −10.74 ± 7.07 mm | 5.48 ± 7.06 mm |
| $X_0$ Gaussian Fit | −10.98 ± 1.78 mm | ± 1.89 mm |

It is a future task to upgrade the LDBS system and increase its accuracy and shorten the acquisition time.

## 9. Conclusions and Outlook

Based on the growing interest in designing space missions where ion beams are used for asteroid deflection or space debris removal, questions about space magnetic field impact on these ion beams have to be addressed. Within this study, a first attempt was undertaken to measure the deflection of an ion beam generated by an RIT radio frequency ion thruster at a distance of several meters. Due to its low beam divergence, the ion thruster is particularly well-suited for a basic study of beam deflection phenomena.

In this experiment, for generating a controllable magnetic field, a large Helmholtz coil of 1.64 m size was built, qualified, and installed between the thruster and diagnostic system. The thruster beam of charged xenon atoms had to pass this magnetic field and was monitored with a flat field scanning system, the LDBS.

The difference to a classical well-defined narrow and unidirectional ion beam (such as in ion optics or particle accelerators) is that we have mixed ion velocities, a not negligible divergence angle, and even electrons from electron sources or secondary processes.

With this, it was possible to measure a beam deflection of about 5–10 mm at a distance of 3.85 m from the thruster exit for a magnetic field of 236 μT. This reproducible result gives confidence that with the used setup it is possible to measure less pronounced deflections for lower magnetic fields. The measurement accuracy of the LDBS seems not high enough to detect a deflection with the same setup and a magnetic field of 40 μT (in addition to the existing geomagnetic field). It should be noted that the deflection values recorded in this study are in the order of the pointing accuracy needed for the mentioned scenario of larger space debris objects removal. On the other hand, it shows that the STG-ET equipment is able to investigate similar phenomena.

This paper will help to answer questions raised by the EP community about the effects of a magnetic field onto a typical electric thruster beam.

A future study will investigate the deflection of a neutralized beam, which is the usual application for space missions. The neutralizer delivers electrons to the beam with the same amount and opposite charge of current. The beam is then quasi-neutral, but we should still see an impact of a magnetic field onto the beam. We undertook preliminary efforts to add electrons to the RIT ion beam with a thermal electron source but not up to full neutralization. A deflection was still detectable, but a comprehensive investigation will be part of a future report.

**Author Contributions:** Concept and methodology: A.N.; preparation of experiment, measurements, investigation: N.S.M.; writing, original draft preparation: A.N.; scientific comments, organization: A.N. and N.S.M. All authors have read and agreed to the published version of the manuscript.

**Funding:** Part of the thruster diagnostic systems has been designed and built, based on a cooperation with the University of Giessen within the project "LOEWE-Schwerpunkt RITSAT—Raumfahrt Ionenantriebe–Plasmaphysikalische Grundlagen und zukünftige Technologien", which was funded by the Hessen State Ministry of Higher Education, Research and the Arts, Germany.

**Data Availability Statement:** All data were collected and archived at DLR.

**Acknowledgments:** We would like to acknowledge the help of Klaus Hannemann, DLR, in organizing the framework around the experiment in his role of head of department. Very sadly, he passed away during preparation and finalizing of this manuscript.

**Conflicts of Interest:** The authors declare no conflict of interest. The funders had no role in the design of the study; in the collection, analyses, or interpretation of data; in the writing of the manuscript or in the decision to publish the results.

**Abbreviations**

| | |
|---|---|
| AMSL | Above mean sea level |
| CoM | Center of mass |
| EP | Electric propulsion |
| LEO | Low Earth orbit |
| LDBS | Long-distance beam scanner |
| NOAA | National Oceanic and Atmospheric Administration |
| RIT | Radiofrequency ion thruster |
| RPA | Retarding potential analyzer |
| STD | Standard deviation |
| STG-ET | High Vacuum Plume Test Facility Göttingen—Electric Thrusters |
| THR | Thruster |

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
