# Peer review of "Ground-Based Experiment for Electric Propulsion Thruster Plume—Magnetic Field Interaction"

_aerospace, doi:10.3390/aerospace10020117_

Round 1
Reviewer 1 Report
The MS titled “Ground-based Experiment for Electric Propulsion Thruster Plume - Magnetic Field Interaction by authors Andreas Neumann, Nina Sarah Mühlich report the theoretical analysis and experimental test of a beam deflection on the RIT ion beam cause by a magnetic field with a magnetic field strength. The result seems has its application significance, however, in my opinions, the MS is from result to result, which lack of necessary analysis and scientific practice. It should be increase scientific connotation before considering to be published.
1. As we know, the electric particles will be deflected by magnetic field. It is basic physical principles. The MS re-proved this principle by a complicated experimental design. Is it the result beyond the calculation according to the basic principle? If not, why the experiment is performed? If so, why it is different?
2. The electric particles deflected by magnetic field is related with the mass of the particles, but I do not find the information of the used propellent, the results is achieved by Xenon, Argon, Krypton? If the results is given by one of them, can it be verified by another propellent? Do it satisfy with the theoretical analysis
3. We know that the Electric Propulsion Thruster has plume divergences. Although the plume divergences angle of ion thruster are fairly small, it is about 10°. Do it affect the experimental results? It should be estimated
4. The mean free path of particles in plume is also another factor might be considered, do it affect the experimental results?
5. Source and influencing factors of measurement error is also lack necessary discussion
6. There are also some text error in the MS. Such as in Error! Reference source not found.
Reviewer 2 Report
This paper for the first time studied experimentally the background magnetic field effects on an electric propulsion thruster plume for transforming momentum to targets. Overall the paper is novel, sound, and well written, so I recommend its publication.
Minor question: Because the thruster has a neutralizer, I expect the beam be neutralized "gradually" (or usually very quickly in a short distance), if we can still see the deflection, does that mean either the beam cannot be well neutralized within the considered distance, or it is deflected before it is fully neutralized, or somewhere in between? I know this would be a future work as mentioned in the conclusion section, but I suggest the authors at least provide some estimation about the extent of the neutralization of the beam, so readers can have a better understanding about what is happening during the combined processes of neutralization and deflection.
P.S. Several "Error! Reference 230 source not found." should be fixed in the paper.
Round 2
Reviewer 1 Report
Authors has answered all my comment. It can be published as it